# Effects of Physical Activity Interventions on Self-Perceived Health Status among Lung Cancer Patients: Systematic Review and Meta-Analysis

**DOI:** 10.3390/cancers15235610

**Published:** 2023-11-28

**Authors:** Alejandro Barrera-Garcimartín, Miguel Sánchez-Polán, Ana López-Martín, María José Echarri-González, Moisés Marquina, Rubén Barakat, Carlos Cordente-Martínez, Ignacio Refoyo

**Affiliations:** 1Faculty of Physical Activity and Sport Sciences-INEF, Universidad Politécnica de Madrid, 28040 Madrid, Spain; a.barrerag@alumnos.upm.es; 2AFIPE Research Group, Faculty of Physical Activity and Sport Sciences-INEF, Universidad Politécnica de Madrid, 28040 Madrid, Spain; barakatruben@gmail.com; 3Oncology Service, Hospital Universitario Severo Ochoa de Leganés, 28911 Leganés, Spain; almartin@salud.madrid.org (A.L.-M.); mecharrigonzalez@gmail.com (M.J.E.-G.); 4Sports and Training Research Group, Sports Department, Faculty of Physical Activity and Sport Sciences-INEF, Universidad Politécnica de Madrid, 28040 Madrid, Spain; moises.mnieto@upm.es (M.M.); carlos.cordente@upm.es (C.C.-M.); ignacio.refoyo@upm.es (I.R.)

**Keywords:** exercise, physical wellbeing, dyspnoea, different stages, randomised control trial

## Abstract

**Simple Summary:**

This article synthesizes the published scientific literature and covers a lack of evidence related to lung cancer patients, their quality of life and the chance to be improved by the physical activity performance. After including 13 articles and assessing obtained results through five meta-analyses, the findings of the current systematic review can give an insight in how physical activity can improve the quality of life in lung cancer patients independent from the stage or treatment of the disease. Thus, physical activity interventions appear to be effective in improving quality of life, physical functioning and physical wellbeing among lung cancer patients.

**Abstract:**

Patients with lung cancer may experience deterioration in quality of life due to adverse effects caused by their disease and its treatment. Although exercise programs have been shown to improve quality of life in certain stages of the disease, the overall impact on this population is unknown. The objective of this research was to evaluate the effect of physical activity on the self-perception of quality of life, physical wellbeing and dyspnea in lung cancer patients. Thirteen articles were included. Five meta-analyses were performed using the standardized mean difference (SMD) with 95% confidence intervals (CI) to evaluate the target outcomes. Results showed significant differences in quality of life (*p* = 0.01; SMD = 0.43, 95% CI = 0.10, 0.75), physical functioning (*p* = 0.01; SMD = 0.27, 95% CI = 0.06, 0.49) and physical wellbeing (*p* = 0.01; SMD = 0.37, 95% CI = 0.08, 0.67) in favour of participants who have undergone the programme compared to those who have not, without significant differences between the two groups in dyspnea. This study shows how physical activity interventions could have positive effects on physical functioning and physical wellbeing but could also be effective for improving quality of life in patients with lung cancer.

## 1. Introduction

Cancer continues to be one of the main causes of mortality worldwide. This disease is experiencing a rapid increase in both its incidence and its mortality around the world [1]. Lung cancer is the second most common cancer-type and the main universal death cause, with a rate of one in five cancer-related deaths [2]. The majority of lung cancer cases could be categorised into two types, non-small cell lung cancer, approximately 70% to 85% from the total of cases; and small cell lung cancer, comprising nearly of 20% to 25% of the cases. The main difference between the types of lung cancers is growth speed and tumour aggressiveness. At the time of diagnosis, more than 70% of patients have tumours in advanced stages or metastases that are not amenable to undergo surgery [3]. The 5-year survival rate is estimated to be around 15% [3,4].

Lung cancer treatment is established according to the type of lung cancer, its stage and functional capacity of the patient. Therapeutic options for lung cancer include surgery, chemotherapy and targeted therapy [5]. More and more treatments are available for these patients. A growing number of alternatives that have an effect on tumour development are being observed. These new pathways can have their effect on different processes and locations, such as signalling cascades that regulate the redox state, as in the case of aprepitant [6]; in nanocompounds, such as naringin–dextrin, which increases apoptosis and suppresses oxidative stress and inflammation [7]; or by combination of new doses of different treatments, like radiotherapy and immunotherapy, helping tumour regression [8]. Early diagnosis systems and improved treatments help to increase the survival rate in this population [9]. Diagnostic systems are being refined, one tool of particular interest in this field is proteomic technology, which identifies and distinguishes differentially expressed proteins compared to their normal tissue [10].

However, symptoms and adverse effects that involve suffering from cancer and its treatment can negatively impact in physical and psychological wellbeing. Frequent symptoms in patients with lung cancer are fatigue, depression and dyspnea, and can be related to each other. These symptoms can affect the daily activities [11]. Additionally, physical activity levels decrease during the course of the disease, having a negative repercussion on functional capacity [12]. It has been observed that this functional capacity could potentially increase the survival rate in this population [13].

Quality of life (QoL) can also be reduced due to the suffer of these symptoms [14]. QoL is defined as a multidimensional concept that encompasses five types of perceptions, physical, material, social and emotional wellbeing, and personal development [15]. Some studies have explored how QoL could decrease in the early stages with patients who have undergone lung resection surgery [16,17] or with advanced lung cancer patients who are undergoing chemotherapy treatment [18]. Consequently, it is necessary to develop interventions that may improve QoL but also reduce the secondary effects related to lung cancer diagnosis and treatment. 

Physical activity has been shown to improve these adverse effects that are related with cancer and QoL. That is how, for example, physical exercise could effectively improve physical functioning and functional capacity [19]. On the other hand, it was observed that this physical exercise could also attenuate cancer-related symptoms [20,21], as is the case of dyspnea [22], an adverse effect that is associated with a major deterioration of quality of life [23].

Recently, a systematic review and meta-analysis have analyzed the effects of physical exercise on perceived QoL among advanced cancer stages in the palliative care phase [24]. This study concluded that exercise could have positive effects on cancer symptoms and QoL in this population. However, the majority of systematic reviews published were focused on analysing the effects of physical activity interventions in patients with general cancer at different stages of the illness or only in specific treatments for patients with general or (particularly) lung cancer.

It is necessary to address global lung cancer patients to understand better the impact of physical activity interventions on self-perceived QoL from a holistic perspective. This systematic review and meta-analysis aimed: (a) to synthesize the scientific literature to determine the adverse effects of physical activity on quality of life in patients with lung cancer, regardless of their stage or treatment; (b) to evaluate the impact of these interventions on physical wellbeing and perceived dyspnoea in this population. 

From a scientific standpoint, the various sections of this study serve to substantiate the robustness of the methodological process and the credibility of the results obtained. It is crucial to emphasize that a comprehensive breakdown of the analytical methodology is available, along with the attributes and outcomes of each article reviewed. The discussion and conclusions provide a comprehensive overview of the obtained results, along with novel perspectives and prospective avenues for research. This is aimed at furnishing effective alternatives to improve the wellbeing of lung cancer patients.

## 2. Materials and Methods

This systematic review and meta-analysis were based on the Preferred Reporting Items for Systematic Reviews and Meta-Analyses (PRISMA) guidelines [25], registering its protocol in the PROSPERO database (the International Prospective Register of Systematic Reviews) (register number: CRD42023418949).

The Participants/Population, Intervention, Comparison, and Outcomes (PICOS) statement was used to analyze the articles characteristics of the included in this review [25]. 

### 2.1. Participants

Patients diagnosed with lung cancer were considered eligible regardless of the phase of the disease or received treatment type. Other types of cancer patients were excluded for this article. 

### 2.2. Intervention

Interventions that were considered eligible had to necessarily involve physical activity or exercise development along them. The analyzed intervention characteristics were, (i) frequency of weekly sessions frequency; (ii) intensity of session; (iii) intervention duration; (iv) type of physical activity, aerobic training, strengthening, mixed or others; (v) if the physical activity program was supervised by a qualified professional; (vi) session duration; (vii) participation of participants in the program; (viii) if the researchers had undergone a complementary intervention.

### 2.3. Comparison

Non-active patients (due to the result of the randomization, they were included in the usual care/control group, they did not realize the physical activity program) who were enrolled in a physical activity intervention were compared with the patients in the intervention group. 

### 2.4. Outcomes

The scores of validated questionnaires on self-perceived QoL assessed at the end of the intervention were the target outcomes. Quality of life questionnaires that were analyzed and included in this review were the following, European Organisation for Research and Treatment of Cancer, Quality of Life Questionnaire Core-30 (EORTC QLQ-C30) and Functional Assessment of Cancer Therapy—Lung (FACT-L).

The EORTC QLQ-C30 questionnaire assesses the quality of life of cancer patients [26,27]. It is scored in a range between 0 and 100 points. It is divided into three sections, global health state/quality of life, functional scale (physical function, daily activities, emotional role, cognitive and social function) and symptoms scale (fatigue, nausea/vomiting, pain, dyspnea, insomnia, loss of appetite, constipation, diarrhoea and financial difficulties). A higher score on the state/quality of life and functioning indicates an improvement in QoL and a better functional level, while a higher score on the symptoms scale represents a higher probability of having more symptoms/problems. 

The FACT-L questionnaire has two parts, general section (FACT-G) and specific lung cancer section (LCS). FACT-G questions involve four general subscales, physical, social/family, emotional, and functional wellbeing. Otherwise, the LCS questionnaire has a specific subscale to assess lung cancer symptoms, with a score range between 0 and 28. The total score after the FACT-L results of adding the FACT-G and LCS scores, can vary from 0 to 136 [28].

General QoL and physical wellbeing were the main outcomes extracted from selected articles, assessed by the previously described questionnaires. The items in EORTC QLQ-C30 questionnaire were global health, physical functioning and dyspnea. Thus, FACT-L analyzed the factors as global score and physical wellbeing. 

### 2.5. Search Strategy, Article Selection, and Data Extraction

Throughout this systematic review, articles written in English and Spanish, published between January 2010 and June 2023, were searched through the MEDLINE (PubMed), Education Resources Information Center (ERIC), SPORTDiscus, Scopus, Web of Science (WoS) and ClinicalTrials.gov databases. The search terms were the following: English: Quality of life AND lung cancer OR lung neoplasms OR lung tumour AND intervention AND exercise OR physical activity OR fitness OR aerobic training OR strength training OR cardiovascular training AND randomized controlled trials OR rct OR randomised control trials OR randomized clinical trial OR randomized OR randomised clinical trial.Spanish: calidad de vida Y cáncer de pulmón O neoplasias pulmonares O tumor pulmonar Y intervención Y ejercicio O actividad física O fitness O entrenamiento aeróbico O entrenamiento de fuerza O entrenamiento cardiovascular Y ensayo clínico aleatorizado O ensayo controlado aleatorizado O eca O aleatorizado O aleatorio.

Included studies were selected following these inclusion criteria: (a) studies with a randomized clinical trial design; (b) participants with a lung cancer diagnosis; (c) intervention program involving quantifiable physical activity or exercise sessions; (d) studies that evaluated quality of life, physical wellbeing or dyspnea after performing a physical activity program. 

Then, the exclusion criteria were: (a) studies involving other types of cancer patients; (b) absence of a usual care/control group (i.e., the lack of a randomized trial design); (c) nonquantifiable physical activity intervention; (d) no validated questionnaire use related to QoL for lung cancer patients; (e) report scores for other items in comparison to the target outcomes in this review; (f) report the results in a different form than the mean and standard deviation (SD). 

From each study, it was descriptively reported, author, year of publication, country of study development, number of participants (general and per study group), type of cancer diagnosed, type of treatment that the participants underwent during the intervention, intervention characteristics (previously mentioned), analyzed variables and type of questionnaire used to measure the quality of life, and if included a co-intervention.

Article identification was performed by two researchers, first analyzing titles and abstracts along previously described databases. Subsequently, articles were excluded if they did not meet the inclusion criteria, then extracting the full texts of those articles who were initially selected to analyze the information contained in them (screening). Finally, a researcher performed the data extraction, retrieving the data in an Excel sheet to perform further analyses. Data extraction was supervised by two other reviewers, assuming that all the information was reported. 

### 2.6. Quality of Evidence and Risk of Bias

The quality of evidence was assessed using the Grading of Recommendations Assessment, Development and Evaluation statement (GRADE). Selected studies were rated with moderate- or high-quality [29]. To measure the potential risk of bias, the Cochrane Handbook was used, analyzing the following sources of bias, selection, performance, detection, attrition and reporting [30]. For each study, these sources of bias were evaluated as low, unclear or high-risk of bias. 

### 2.7. Statistical Analysis

In this systematic review, five meta-analyses were performed using the review manager software (RevMan, version 5.3, EE UU). All analyses assessed continuous variables as a result of a validated questionnaire. The first and second meta-analysis included articles that recovered quality of life scores: global health corresponding to the EORTC QLQ-C30 questionnaire and the global score of the FACT-L questionnaire. The next meta-analyses aimed to analyze physical well-being: corresponding to the physical functioning item in the EORTC QLQ-C30 questionnaire and physical well-being in the FACT-L questionnaire. Finally, a meta-analysis was performed to assess the dyspnoea using the dyspnoea specific item of the EORTC QLQ-C30 questionnaire.

For all analyses, the mean and standard deviation were recovered to compare the result after the intervention between the groups, performing and inverse variances (IV) using standardized mean differences (SMD) analysis, using the Hedges G formula to take into account the effects’ sizes per article. The random effects model was used to report overall 95% confidence intervals (CI) [31]. 

A relative weight system was assigned to each study depending on the sample size, calculating the compensated average of each research to specifically attribute the weight to the general analysis. To determine whether the results were statistically significant, a *p*-value of 0.05 was set. The interpretation of the effect sizes was performed considering that a value lower than 0.4 indicated a small effect, between 0.4 and 0–7 was considered a moderate effect and more than 0.75 was a high effect [32]. To calculate heterogeneity, the I^2^ statistic was used determining the percentage of variability in each analysis (between articles), taking into account the following ranges, low (25%), moderate (50%) or high (75%) heterogeneity [33]. An approach used to address high-heterogeneity involves putting the studies into subgroups based on certain descriptive characteristics that could explain the variability between them [34]. Despite the fact that high-heterogeneity was reported in one meta-analysis, it was decided to not split the articles into subgroups due to the low number of articles included in this analysis, considering that this approach provides a more complete view of the study. 

## 3. Results

After the initial search, a total of 251 articles were identified. Thus, 47 articles were excluded because of deduplication having 204 reports. Consequently, 159 articles were excluded after reviewing their titles and abstracts because they did not meet the previously established inclusion criteria. 45 articles were assessed for eligibility, analysing full-texts of them, finally excluding 32 articles for various reasons: inclusion in the research patients with different types of cancer (n = 14), did not use validated questionnaires for this type of population or did not report the results of the items analyzed (n = 8), did not indicate the mean values or standard deviation in the questionnaire scores (n = 6), the intervention program was not considered as physical activity or exercise (n = 4). Then, 13 randomized clinical trials [35,36,37,38,39,40,41,42,43,44,45,46,47] were analyzed in the review (Figure 1). 

### 3.1. Description of Participants Characteristics 

The articles involved a total of 628 participants. There was a large variability in the diagnosis of participants along articles that were included in this review (Table 1). Some studies were carried out with patients diagnosed with non-small cell lung cancer involving all phases, from stage I to stage IV [35,38,44,45]. Other studies enrolled patients in early stages of the disease, from stage I to stage IIIa [37,40,42,46], or in more advanced stages, from stage IIIa to stage IV [36,39,41,43,47].

It was possible to distinguish different periods in which the intervention was performed and several types of treatment. Intervention moments compromised the period before surgery [40,42,44,45] and after surgery [35,37,38,46]. Regarding treatment, the studies involved patients with chemotherapy [37,41,47], concomitant chemoradiotherapy [39], targeted therapy [43] or with other different types of treatment types [36].

### 3.2. Intervention Characterisctics

Table 1 shows intervention and general participants and articles characteristics. In the selected articles, all participants belonged to intervention group carried out a physical activity or physical exercise program. There was a large variability in the duration of the program in included studies, with interventions lasting from one week [42,44,45] up to twenty weeks [38]. At the same time, both the weekly frequency of the sessions and their duration presented great diversity, ranging from 90 min of training sessions twice a week [47], to daily training of 15 to 30 min per session [44].

Regarding the type of physical activity, most of the studies mainly performed aerobic training independently [39,43] or mixed with other types of training, such as strengthening [41,47], resistance training [37,38,40] or breathing exercises [42,44,45,46]. On the other hand, there were physical activity interventions that involved activities such as walking [36] or been combined with strength exercises [35]. The intervention program intensity was majorly high [38,42,44,45], reaching very-high-intensity in two studies [43,47]. In the remaining investigations, four involved moderate intensity [35,40,41,46], two studies combined moderate- and high-intensity [37,39] and one article established a low intensity throughout its intervention [36]. 

Mostly, the sessions were supervised by a professional [37,38,39,41,42,43,44,45,46,47]. In two articles, a program in which part of the intervention was supervised while the other was not was implemented [35,40]. In one article, any professional supervised the sessions [36]. Only one article performed a complementary (co-intervention) intervention with the physical activity program [40], carrying out also a nutrition program. 

Nine studies used the EORTC QLQ-C30 questionnaire as a measurement tool for quality of life [35,36,38,41,42,43,44,45,46], three works were developed with the FACT-L questionnaire [39,40,47] and only one article involved both questionnaires [37].

### 3.3. Risk of Bias and Quality of Evidence Assessments

Risk of bias assessment is shown in Figure 2. Generally, risk of bias in the participants selection was low due to the randomized trial design of all the selected articles. A higher risk in performance and detection sources of bias due to the nature of these interventions and due to the difficulty to blind the participants in both the intervention and control group was observed. Regarding attrition source of bias, it is important to take into account the situation to which its population is exposed and its greater risk of abandonment. Finally, it was difficult to access to some protocols of retrieved articles, supposing a higher risk of bias in reporting bias. Overall, the quality of evidence resulted in a high level of evidence also with high-importance because of the design (randomized clinical trial) of the articles that were selected.

### 3.4. Physical Activity Effects on Self-Perceived Quality of Life

#### 3.4.1. Analysis of EORTC QLQ-C30 Questionnaire

Nine studies [35,36,37,41,42,43,44,45,46] were included to analyze quality of life using the EORTC QLQ-C30 questionnaire. Figure 3 shows the effect of physical activity on global health status/quality of life among the included interventions, showing a statistically significant effect with an improvement in those participants who underwent a physical activity intervention compared to those who didn’t perform it (Z ₌ 2.54, *p* = 0.01; SMD ₌ 0.43, 95% CI ₌ 0.10, 0.75, I^2^ ₌ 58%, P_heterogeneity_ ₌ 0.01). 

#### 3.4.2. Assessment of FACT-L Questionnaire

However, no significant differences between intervention and control groups in the four included studies [37,39,40,47] about global score of FACT-L questionnaire (Z ₌ 1.74; *p* < 0.08; SMD ₌ 0.29, 95% CI ₌ −0.04, 0.62, I^2^ ₌ 7%, P_heterogeneity_ ₌ 0.36) (Figure 4).

### 3.5. Physical Activity Effects on Self-Perceived Wellbeing

#### 3.5.1. Analysis of EORTC QLQ-C30 Questionnaire

Seven randomized clinical trials [36,41,42,43,44,45,46] were included in this analysis, retrieving physical functioning (Figure 5) showing a significant increase in the intervention group compared to the participants of the control group (Z ₌ 2.48; *p* = 0.01; SMD ₌ 0.27, 95% CI ₌ 0.06, 0.49, I^2^ ₌ 0%, P_heterogeneity_ ₌ 0.49).

#### 3.5.2. Assessment of FACT-L Questionnaire

This meta-analysis integrated four studies [37,39,40,47] with respect to physical wellbeing, obtaining a significant positive effect in those participants who belonged to the intervention group compared to non-active participants (Z ₌ 2.51; *p* = 0.01; SMD ₌ 0.37, 95% CI ₌ 0.08, 0.67, I^2^ ₌ 0%, P_heterogeneity_ ₌ 0.46) (Figure 6).

### 3.6. Physical Activity Effects on Self-Perceived Dyspnea 

In this analysis, four articles [36,38,41,46] were analyzed reporting dyspnea scores (Figure 7). No significant differences were observed between groups regarding this factor (Z ₌ 1.68, *p* = 0.09; SMD ₌ −0.83, 95% CI ₌ −1.79, 0.14, I^2^ ₌ 86%, P_heterogeneity_ ₌ 0.00).

## 4. Discussion

The purpose of this systematic review and meta-analysis was to analyze the effect of physical activity interventions on the self-perceived quality of life among lung cancer patients. Secondary objectives compromised to assess this impact on perceived physical wellbeing and dyspnea in the same population. The main result obtained indicates that physical activity programs have positive effects on perceived physical wellbeing and moderate effect size on self-perceived quality of life. 

This research implemented a wide variety of inclusion criteria to ensure greater control over the effect of physical activity interventions on the analyzed outcomes. Several meta-analyses performed in this area have analyzed the scores of the studied outcomes combining QoL questionnaires for both the general population and specific questionnaires for cancer patients. This article is worth highlighting due to the use and analysis of validated questionnaires that are specifically for this population. Demonstrating the differences between the QoL questionnaires, a study evaluated the correlation between the SF-36 health questionnaire and the EORTC QLQ-C30 questionnaire. The results revealed a low correlation between the most generic questionnaire (SF-36) and specific questionnaires that are specifically designed for people who suffer from this disease [48]. 

Regarding physical wellbeing, both questionnaires that were retrieved in this review showed significant benefits to the participants of the intervention group compared to the control group, with a small size effect. These results are in line with the results of two previously published meta-analyses related to the postsurgical phase. The study by Machado et al. [49] which integrated seven articles and a second investigation in which four articles were included [50], both observed positive effects on the physical functioning element from the quality of life questionnaire in patients with lung cancer in the intervention group after surgery. However, a meta-analysis conducted with three studies that included patients in an advanced stage did not yield significant results [51]. It should be noted that these studies included interchangeably different questionnaires. According to the study developed by Machado et al. [49], improvements in physical appearance were found to be less significant when using cancer-specific questionnaires compared to generic instruments. Therefore, the results obtained in this meta-analysis acquire greater relevance, highlighting the importance these interventions have on this parameter. 

The importance of physical activity in this population should not be ignored, in which patients with non-small cell lung cancer have been shown to have a reduced level of physical activity at the time of diagnosis and are exposed to greater risks of mortality [52]. The impact of physical activity has an impact on general wellbeing after surgery, with a significant relationship between a higher level of physical activity before an oncological surgery and a better quality of life after this intervention [53]. Preserving physical function through exercise as the disease progresses could play an essential role in improving quality of life [54]. This is what patients diagnosed with lung cancer have expressed, with a reduction in quality of life and functional capacity being one of their main concerns [55]. 

For all these reasons, improving physical functioning becomes crucial in this population. The meta-analyses carried out in this review have had a great variety in terms of the phase in which the participants were, despite this, heterogeneity observed in them was low. The analysis of the EORTC QLQ-C30 questionnaire has covered articles that include patients in advanced stages [36,41,43], early stages [42,46], and integrating all types of stages [44,45]. The FACT-L questionnaire included two articles with patients in early stages [37,40] and two research studies with participants in advanced stages [39,47]. Among all the included works, different types of treatment were observed, surgery, chemotherapy, targeted therapy, etc. Therefore, through these meta-analyses, it has been shown that exercise interventions are effective in positively improving the perceived physical wellbeing in patients with lung cancer, regardless of their stage or type of treatment received. 

The effect of physical activity on self-perceived quality of life in the EORTC QLQ-C30 questionnaire showed a moderate effect size. Consistent with these results, Cavalheri et al. [56] observed significant effects in the intervention group with patients over the next 12 months after lung resection. Similarly, two meta-analyses reported significant differences in favour of participants who performed the physical activity intervention, both in patients in the palliative care phase [24] and in those with advanced lung cancer [51]. On the other hand, regarding the FACT-L questionnaire, our analysis did not obtain a significant difference between the study groups, as did the meta-analysis carried out with patients with lung cancer after surgery [49]. Adjuvant therapy has been associated with a decrease in the physical aspect of quality of life [57], in which it has been observed how chemotherapy treatment can minimize the positive effect of interventions with post-surgery lung cancer patients [58]. This could be one of the reasons why no significant differences were found in the analysis of the FACT-L questionnaire, since in three of the four articles included in the study, patients were receiving chemotherapy treatment. 

It is important to point out the symptoms and adverse effects to which these patients were exposed and which can significantly influence their quality of life. Patients may suffer long-term side effects such as fatigue, pain, insomnia [59] and muscle wasting [60] due to treatments such as chemotherapy or radiotherapy. Surgical intervention to treat lung cancer is associated with complications, functional limitations, and deterioration of quality of life [61], in which it can take up to two years to reach their level of global health prior to the operation [62]. Patient response to each treatment is individual and can be affected in different ways through different factors such as the toxicity of the treatment or its impact on organ function. Consequently, any improvement that occurs, no matter how minimal, can help considerably to restore or increase the QoL of this population. Although, not having obtained significant results in one of the meta-analyses, it is necessary to highlight the significant difference and the size of the effect obtained in the EORTC QLQ-C30 questionnaire. Nine articles were included in it, of which three included patients in different stages of the disease [35,44,45], three included patients in early stages [37,42,46] and another three involved patients in advanced stages [36,41,43]. Different treatment modalities were observed among all included investigations. It appears that physical activity interventions may be a beneficial tool in improving the quality of life of patients with lung cancer. 

Through semi structured interviews, a study revealed that the majority of patients who had undergone pulmonary lobectomy declared that they were concerned about dyspnea [63]. Dyspnea can hinder participation in physical and daily activities, resulting in decreased levels of physical activity [12], affecting autonomy and independence. It has been shown that physical exercise helps reduce side effects and can be even more effective than drug treatment [64]. In cancer survivors, physical exercise has been shown to provide benefits on several of the symptoms of the disease and for the side effects derived from treatments [65]. An example of this is fatigue, where treatment-related fatigue can transform into exercise-induced fatigue, which is perceived more positively as a result of exercise participation [66]. 

However, the analysis of dyspnea carried out did not report significant differences between groups and presented high-heterogeneity. A previous study that included two articles with patients during the year following lung resection surgery showed a decrease in dyspnea symptoms [56]. In contrast, a study that analyzed five articles with patients with advanced lung cancer did not observe significant differences [51]. Due to the lack of studies in the first mentioned research, the use of different measurement instruments in this last study and the scarcity of articles, as well as the high-heterogeneity in the meta-analysis carried out in this research, there is not enough evidence to be able to draw conclusions on this parameter. 

Most oncologists consider exercise to be beneficial and important. Despite this, more than half of them have the perception that it is not safe and only 7.2% of them believe that cancer patients manage to exercise during cancer treatment [67]. Based on our findings, the results led us to determine the importance of performing physical activity interventions in patients with lung cancer as an effective strategy to improve their physical wellbeing and perceived quality of life regardless of the stage or type of treatment they are undergoing.

It is essential to analyze the factors present in physical activity programs that may be related to the improvement of the analyzed variables. A key element could be the supervision of the sessions. Most of the included studies supervised the sessions, only one investigation did not carry out any type of supervision [36]. Home exercise programs can help promote exercise [68]; however, it appears that supervised group training helps generate greater adherence and social support, providing a number of additional benefits to participants, including in the oncology population with a poor prognosis and symptomatology [69,70]. 

Due to the limited number of articles included, the diversity of intervention programs and the different characteristics of the participants, we cannot provide specific recommendations on the exercise prescription necessary to improve quality of life, physical wellbeing and perceived dyspnea in patients with lung cancer. However, it is important to highlight that aerobic training was the most frequently selected form of exercise in the programs, with high-intensity being the most widespread option among the interventions. There are other variables that can also influence the effectiveness of these interventions such as the duration of the program, frequency and adherence to exercise [71]. Nevertheless, more studies are needed to analyze the effect of exercise on quality of life to be able to respond to the research topic raised.

It is important to point out the limitations of the present systematic review and meta-analysis to achieve a better understanding of the results. The lack of consensus on the quality of life scales used in this population represents the main limitation of this study. This discrepancy has resulted in a smaller number of eligible articles and the need to form subgroups based on the different questionnaires used. The scarcity of literature in this area has had an impact on the heterogeneity of exercise programs carried out, making it difficult to obtain practical conclusions about exercise prescription. This research has only assessed the scores after the intervention, without considering the initial values of each participant. Furthermore, other variables that could influence the quality of life of participants, such as age or sex, have not been considered. Consequently, future studies should include the same measurement scales to obtain a more complete view. 

## 5. Conclusions

This systematic review synthesized the existing literature on physical activity and its effect on quality of life, physical wellbeing and dyspnea. The results of this study show the effectiveness of physical activity interventions in generating improvements in perceived physical wellbeing and appear to be an effective strategy to improve perceived quality of life in patients with lung cancer, regardless of the stage of the disease or the type of treatment they are receiving. However, it has not been possible to draw any conclusions about the impact of physical activity programs on the perception of dyspnea in this population.

The data from this study suggest that physical activity programs could benefit patients with lung cancer. However, this study had some limitations, including a restricted inclusion of articles, varied intervention programs and diverse participant characteristics. Future studies in this field should investigate which variables of the training programme might have the greatest impact on quality of life, physical wellbeing and dyspnea.

## Figures and Tables

**Figure 1 cancers-15-05610-f001:**
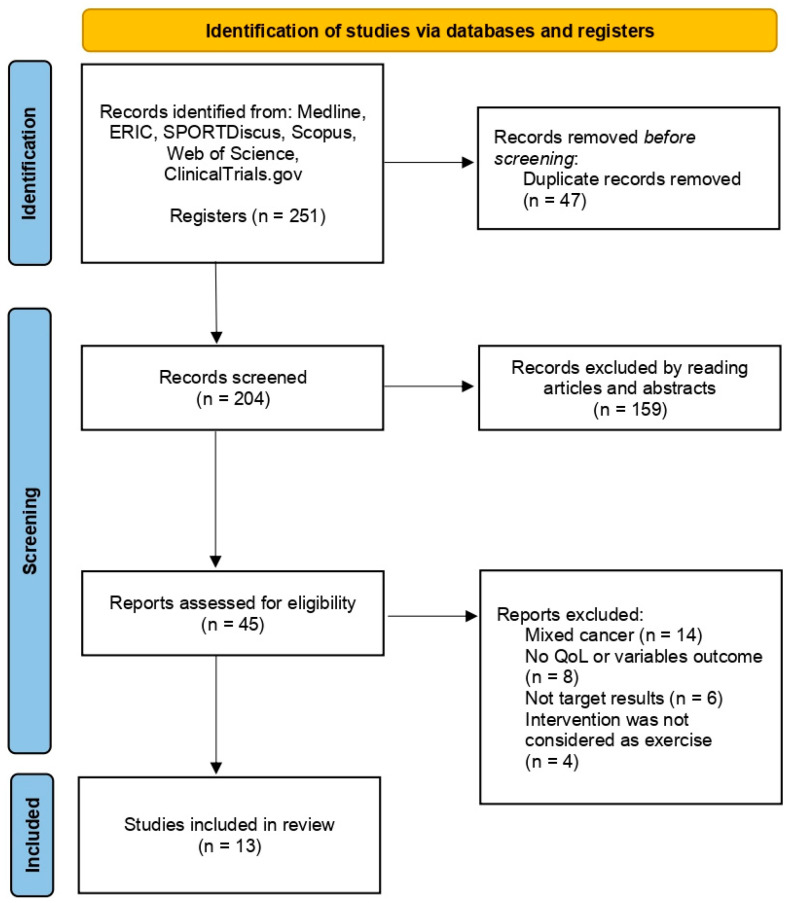
Article selection process [35,36,37,38,39,40,41,42,43,44,45,46,47].

**Figure 2 cancers-15-05610-f002:**
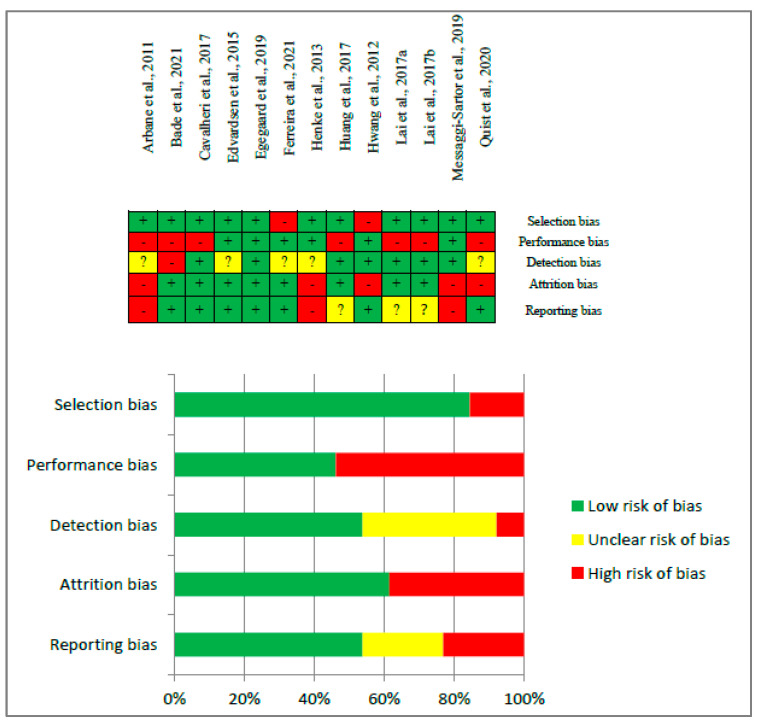
Risk of bias assessment [35,36,37,38,39,40,41,42,43,44,45,46,47].

**Figure 3 cancers-15-05610-f003:**
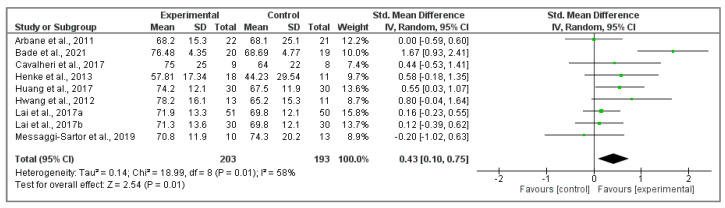
Physical activity effects on global health state with EORTC QLQ-C30 questionnaire [35,36,37,41,42,43,44,45,46].

**Figure 4 cancers-15-05610-f004:**
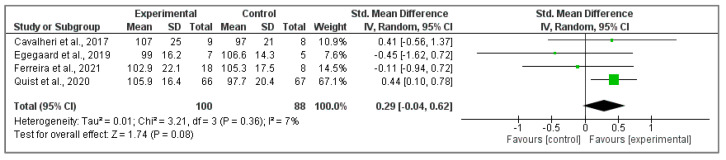
Effects of physical activity on global score of FACT-L questionnaire [37,39,40,47].

**Figure 5 cancers-15-05610-f005:**
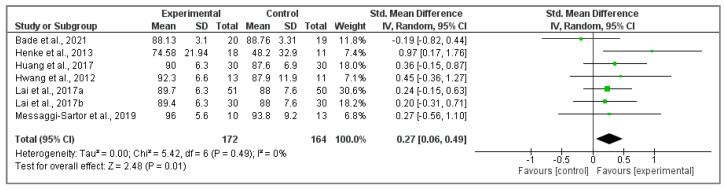
Effects of physical activity interventions on physical functioning EORTC QLQ-C30 questionnaire score [36,41,42,43,44,45,46].

**Figure 6 cancers-15-05610-f006:**
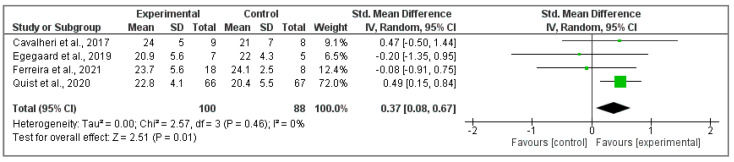
Physical activity effects in physical wellbeing of FACT-L questionnaire score [37,39,41,47].

**Figure 7 cancers-15-05610-f007:**
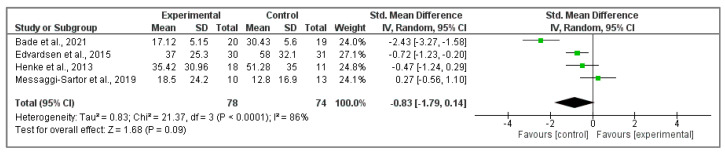
Effect of physical activity on dyspnea EORTC QLQ-C30 questionnaire score [36,38,41,46].

**Table 1 cancers-15-05610-t001:** Article characteristics.

Author, Year and Country	N; IG; CG	Diagnosis	Treatment		Intervention. Physical Activity Program						Main Variables Analyzed. QoL Assessment Tool	Co Intervention
				W Frec.	Int.	Time	Type	Sup.	Duration	Adh.		
Arbane et al., 2011. United Kingdom [35]	43; 22; 21	Stage I–V NSCLC	Post-surgery	5 days: twice daily	Mod	5 days	Strength and mobility training	Sup.	ND	ND	Muscle strength, QoL and exercise tolerance. EORTC QLQ-C30	No
				ND	ND	12 weeks	Strength training and walking	Unsup.	ND			
Bade et al., 2021. United States [36]	39; 20; 19	Stage III or IV NSCLC	Immunotherapy, chemotherapy, targeted therapy, post-treatment	7 days	Low	12 weeks	Walking	Unsup.	ND	ND	Physical activity, QoL, dyspnoea, depression and biomarkers. EORTC QLQ-C30	No
Cavalheri et al., 2017. Australia [37]	17; 9; 8	Stages I–IIIA NSCLC	Postoperative chemotherapy or after lobectomy	3 days	Mod, high	8 weeks	Aerobic and resistance training	Sup.	60 min	44%	Exercise capacity, physical activity, peripheral muscle force, QoL, fatigue, anxiety, depression and lung function. FACT-L; EORTC QLQ-C30	No
Edvardsen et al., 2015. Norway [38]	61; 30; 31	Stage I–IV NSCLC	Post-surgery	3 days	High	20 weeks	Interval, resistance and inspiratory muscle training	Sup.	60 min	88%	Peak oxygen uptake, pulmonary function, muscle strength, total muscle mass, daily physical functioning and QoL. EORTC QLQ-C30	No
Egegaard et al., 2019. Denmark [39]	12; 7; 5	Stage IIIa–IV NSCLC	Concomitant chemoradiotherapy	5 days	Mod, high	7 weeks	Aerobic interval training	Sup.	20 min	88%	VO2 peak, functional capacity, pulmonary function, anxiety, depression and QoL. FACT-L	No
Ferreira et al., 2021. Canada [40]	26; 18; 8	Lung cancer stages I, II or IIIa	Pre-surgery	1 day	Mod	4 weeks	Aerobic and resistance training	Sup.	60 min	82%	Functional capacity, QoL, anxiety, depression, energy expenditure, nutritional status, body composition, physical function, anaerobic threshold, V02 peak. FACT-L	Nutritio-nal supplement
				6 days	Mod		Aerobic, resistance and stretching training	Unsup.	35 min			
Henke et al., 2013. Germany [41]	29; 18; 11	Stage IIIA/IIIB/ IV NSCLC or SCLC	Palliative platinum-based chemotherapy	6 days	Mod	12–14 weeks	Endurance and strength training and breathing techniques	Sup.	ND	>75%	Barthel Index, QoL, endurance capacity, dyspnoea perception and muscle strength. EORTC QLQ-C30	No
Huang et al., 2017. China [42]	60; 30; 30	Stage I–III NSCLC	Pre-surgery	7 days	High	1 week	Aerobic and inspiratory muscle training	Sup.	20–40 min	90%	Postoperative pulmonary complications, length of hospital stay, QoL, functional capacity and peak expiratory flow. EORTC QLQ-C30	No
Hwang et al., 2012. Taiwan [43]	24; 13; 11	Stage IIIA–IV NSCLC	Targeted therapy	3 days	High, very high	8 weeks	Aerobic interval training	Sup.	30–40 min	71%	VO2peak, muscle strength, endurance and oxygenation during exercise, insulin resistance, inflammatory response and QoL. EORTC QLQ-C30	No
Lai et al., 2017a. China [44]	101; 51; 50	Stage I–IV NSCLC	Pre-surgery	7 days	High	1 week	Aerobic training. Thoracic expansion and breathing exercises.	Sup.	15–30 min	ND	Postoperative pulmonary complications, QoL, functional capacity, changes in blood gas. EORTC QLQ-C30	No
Lai et al., 2017b. China [45]	60; 30; 30	Stage I–IV NSCLC	Pre-surgery	7 days	High	1 week	Aerobic and inspiratory muscle training	Sup.	30 min	ND	Postoperative pulmonary complications, QoL, functional capacity and peak expiratory flow. EORTC QLQ-C30	No
Messaggi-Sartor et al., 2019. Spain [46]	23; 10; 13	Stage I or II NSCLC	Post-surgery	3 days	Mod	8 weeks	Aerobic and inspiratory and expiratory muscle training	Sup.	60 min	80%	Exercise capacity, respiratory muscle strength, QoL, levels of serum insulin growth factor I (IGF-I) and IGF binding protein 3 (IGFBP-3). EORTC QLQ-C30	No
Quist et al., 2020. Denmark [47]	133; 66; 67	Stage III or IV NSCLC or ED-SCLC	Chemotherapy	2 days	High, very high	12 weeks	Strength, aerobic, stretching and relaxation exercises	Sup.	90 min	44%	V02 peak, muscle strength, functional capacity, pulmonary function, QoL, anxiety and depression. FACT-L	No

IG, intervention group; CG, control group; W freq., weekly frequency; Int., intensity; Adh., adherence; QoL, quality of life; NSCLC, non-small cell lung cancer; SCLC, small cell lung cancer; ED-SCLC, extensive-disease small-cell lung carcinoma; Mod, moderate; Sup., supervised sessions; Unsup., unsupervised sessions; ND, No data available; min, minutes; EORTC QLQ-C30, European Organization for Research and Treatment of Cancer, Quality of Life Questionnaire Core-30; FACT-L, Functional Assessment of Cancer Therapy—Lung.

## Data Availability

The data presented in this study are available on request from the corresponding author.

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
