# Peer review of "Effects of Physical Activity Interventions on Self-Perceived Health Status among Lung Cancer Patients: Systematic Review and Meta-Analysis"

_cancers, 2023, doi:10.3390/cancers15235610_

Round 1
Reviewer 1 Report
Comments and Suggestions for Authors
22 October 2023
Ms. Ref. No.: cancers-2682776
Journal: Cancers
Title: Effects of physical activity interventions on self-perceived health status among lung cancer patients: systematic review and meta-analysis
Comments:
Thank you for your efforts in writing this article on a very pertinent topic. Moreover, I found the article to be informative and with the potential for further research on this topic in future.
I have some observations where mentioned in the following paragraphs that will be useful for its improvement:
1- The document contains some areas that could benefit from improvement.Two figures are marked with the same number, which might cause confusion and should be corrected.
2- Clarify the meaning of the 13 articles mentioned in the methods section. This will make it easier for the reader to understand the methodology and the rationale behind the research.
3- Additionally, while the document mentions that there is no limit on the year of publication of articles, it is important to note that using recent studies and articles can improve the validity and reliability of the research in question.
4- Moreover, The Following reference can be included in the introduction part for more readability:
· https://doi.org/10.3390/cancers15205102
· https://doi.org/10.1155/2022/8540403
· https://doi.org/10.3390/cancers15205083
· https://doi.org/10.1177/02601060221082384
· https://doi.org/10.3390/cancers15205005
5- Finally, it is again stated that some effects during the treatment of patients create limitations in improving their quality of life! Provide more details on the limitations of the treatment of patients and the factors that affect their quality of life. This will help the reader to better understand the implications of the research and the potential for future studies to address these limitations
Author Response
Thank you very much for taking the time to review this manuscript. We really appreciate your recommendations. Please find the detailed responses below and the corresponding corrections in track changes in the re-submitted files.
Comment 1: The document contains some areas that could benefit from improvement. Two figures are marked with the same number, which might cause confusion and should be corrected.
Thank you for pointing this out. We have changed the number of figures correctly.
Comment 2: Clarify the meaning of the 13 articles mentioned in the methods section. This will make it easier for the reader to understand the methodology and the rationale behind the research.
We agree with this comment. Therefore, the information on the selection of articles has been extended. The modification can be found: page number 5, results section and lines 236-240.
Comment 3: Additionally, while the document mentions that there is no limit on the year of publication of articles, it is important to note that using recent studies and articles can improve the validity and reliability of the research in question.
We really appreciate your observations. We have, accordingly, modified the limitation of the year of publication of the articles to emphasize this point. Changes can be observed: page 4, section 2.5 and lines: 158 and 172.
Comment 4: Moreover, The Following reference can be included in the introduction part for more readability.
Thank you very much for taking the time to provide us with such innovative and specific articles. Most of the recommended studies have been included. The included references can be found: page 2, 2nd paragraph and lines: 55-65.
Comment 5: Finally, it is again stated that some effects during the treatment of patients create limitations in improving their quality of life! Provide more details on the limitations of the treatment of patients and the factors that affect their quality of life. This will help the reader to better understand the implications of the research and the potential for future studies to address these limitations.
We concur with the suggestion. We have added some references to support this idea. The changes can be seen: page 14, 4th paragraph and lines: 399-401and 404-406; page 15, 1st paragraph and lines: 418-420.
Reviewer 2 Report
Comments and Suggestions for Authors
Reviewer’s Report on the manuscript entitled:
Effects of physical activity interventions on self-perceived health status among lung cancer patients: systematic review and meta-analysis
The authors provided a systematic review for evaluating the effect of physical activity on the self-perception of quality of life, physical wellbeing, and dyspnea in lung cancer patients. Generally, the manuscript is well-written. In my view, however, physical activity and mild workout have always been the key for wellbeing and better QoL, making this research not significant. Below, please see my comments.
In the Simple Summary where you wrote:
“…systematic review could enhance physical activity in lung cancer….”. This sentence is meaningless because your review can not enhance physical activity! I suggest replacing “could enhance” with “can give an insight in how physical activity can improve the quality of life in lung cancer patients.” Or something like that.
You wrote several statistical metrics without defining the acronyms in the Abstract, such as SMD, CI, etc. In addition, this listing these values are not clear for general readers, and they will not be able to understand your work. Therefore, I suggest re-writing your abstract. Please mention the objectives, main findings, etc.
At the end of Introduction, please list how the rest of this manuscript is organized.
The conclusion section is poorly written. Please briefly summarize the objectives, search methods, findings, limitations, and recommendations in this section.
Other minor editorial comments:
There are many grammar/typo/punctuation issues in the manuscript that should be checked and corrected. For example, I mention below a few, but there are more:
Section 2.7. Grammar issue: “…were recovered to compared the…”
Page 10 and Page 11. Figure 2 appeared twice. The one on the top of Page 11 should be Figure 3, then Figure 4, etc. Please also improve the quality and readability of these figures.
Thank you!
Comments on the Quality of English LanguageThere are many grammar/typo/punctuation issues in the manuscript that should be checked and corrected.
Author Response
Thank you very much for taking the time to review this manuscript. We really appreciate your recommendations. Please find the detailed responses below and the corresponding corrections in track changes in the re-submitted files.
Comment 1: In the Simple Summary where you wrote: “…systematic review could enhance physical activity in lung cancer….”. This sentence is meaningless because your review can not enhance physical activity! I suggest replacing “could enhance” with “can give an insight in how physical activity can improve the quality of life in lung cancer patients.” Or something like that.
Thank you very much for the appreciation. We have done the modification. Change can be observed: page 1, lines: 20-21.
Comment 2: You wrote several statistical metrics without defining the acronyms in the Abstract, such as SMD, CI, etc. In addition, this listing these values are not clear for general readers, and they will not be able to understand your work. Therefore, I suggest re-writing your abstract. Please mention the objectives, main findings, etc.
We agree with this comment. Therefore, we have re-written the abstract. The changes can be found: page 1, lines: 30-37.
Comment 3: At the end of Introduction, please list how the rest of this manuscript is organized.
Thank you very much for your recommendation. At the end of the introduction, we have added the organisation of the manuscript: page 3, lines: 100-106.
Comment 4: The conclusion section is poorly written. Please briefly summarize the objectives, search methods, findings, limitations, and recommendations in this section.
We concur with the suggestion. We have tried to expand the information in this section with the proposed suggestions. The modifications can be seen: page 16, lines: 472-484.
Comment 5: There are many grammar/typo/punctuation issues in the manuscript that should be checked and corrected. For example, I mention below a few, but there are more. Section 2.7. Grammar issue: “…were recovered to compared the…”
Agree. We have, accordingly, revised and corrected the grammar issues of the manuscript. The suggested correction has been modified: page 5, 2nd paragraph, line: 212.
Comment 6: Page 10 and Page 11. Figure 2 appeared twice. The one on the top of Page 11 should be Figure 3, then Figure 4, etc. Please also improve the quality and readability of these figures.
Thank you for pointing this out. We have changed the number of figures correctly. We have also tried to improve the image quality of the graphics. We can still improve it further, but the file would be too heavy. However, we have attached the images to the editor with better image quality.
Reviewer 3 Report
Comments and Suggestions for Authors
This metaanalysis aimed to demonstrate a positive effect of physical activity among patients affected by lung cancer.
The study is well conducted and the paper is well written.
I would not include studies in Spanish. This might be a selection bias.
Selection of articles must be further detailed.
Author Response
Thank you very much for taking the time to review this manuscript. We really appreciate your recommendations. Please find the detailed responses below and the corresponding corrections in track changes in the re-submitted files.
Comment 1: I would not include studies in Spanish. This might be a selection bias.
Thank you very much for your recommendation. We conducted the search in Spanish to be able to expand the search and include more articles. We considered that by broadening the search to more languages, the article could gain a greater richness, as long as they met the inclusion and exclusion criteria and did not present a high risk of bias. We will take this suggestion into account for future research.
Comment 2: Selection of articles must be further detailed.
We agree with this comment. Therefore, the information on the selection of articles has been extended. Changes can be found: page number 5, results section and lines 236-240.
Round 2
Reviewer 2 Report
Comments and Suggestions for Authors
Dear authors,
Thank you for addressing my comments and improving your manuscript.
Comments on the Quality of English LanguageThere are some typos/grammar/punctuation issues that can be fixed.